# Feature-Based Interpretation of the Deep Neural Network

Eun-Hun Lee  and Hyeoncheol Kim *

Department of Computer Science and Engineering, Korea University, 145 Anam-ro, Seongbuk-gu, Seoul 02841, Korea; booksky@korea.ac.kr
* Correspondence: harrykim@korea.ac.kr

**Abstract:** The significant advantage of deep neural networks is that the upper layer can capture the high-level features of data based on the information acquired from the lower layer by stacking layers deeply. Since it is challenging to interpret what knowledge the neural network has learned, various studies for explaining neural networks have emerged to overcome this problem. However, these studies generate the local explanation of a single instance rather than providing a generalized global interpretation of the neural network model itself. To overcome such drawbacks of the previous approaches, we propose the global interpretation method for the deep neural network through features of the model. We first analyzed the relationship between the input and hidden layers to represent the high-level features of the model, then interpreted the decision-making process of neural networks through high-level features. In addition, we applied network pruning techniques to make concise explanations and analyzed the effect of layer complexity on interpretability. We present experiments on the proposed approach using three different datasets and show that our approach could generate global explanations on deep neural network models with high accuracy and fidelity.

**Keywords:** neural network; explainable artificial intelligence (XAI); interpretability

## 1. Introduction

Unlike conventional machine learning techniques, which perform well while trained with hand-designed features extracted by humans, deep neural network models show decent performance even using low-level data directly because units in the upper layer can represent high-level features by information acquired from the lower layer [1]. Based on the characteristic that neural networks map low-level features to high-level features during the training process, transfer learning has been proposed to reduce the training cost of neural networks [2]. The basic idea of transfer learning is to import the network parameters from a trained model with a similar data domain. This method makes it possible to skip the process of training high-level features from low-level data and build a new neural network from high-level features suitable for the desired application.

However, it was not easy to guarantee that the neural network acquired sufficient knowledge about the domain during the training process. Contrary to expectations, deep neural networks only have a shallow understanding of a specific task and thus have a limited capacity to transfer knowledge [3]. Determining whether a neural network has captured sufficient high-level features from the data domain is essential to avoid adapting transfer learning to model with unsuitable data.

Various explainable artificial intelligence (XAI) studies explain the decision-making process of neural network models in the human-understandable form to solve the difficulty of intuitively understanding acquired knowledge of the model [4–6]. Classical approaches have provided explanations in logical rules from combinations of input units that activate output units. These approaches could provide adequate explanations with structured data but produced explanations that were difficult for humans to understand for unstructured data, such as images. Modern approaches, such as CAM [7], LIME [8], and

attention-based explanation [9], provide a local explanation for unstructured data. However, these approaches only generate an explanation for a single instance and only indicate the area the neural network focuses on rather than represent the acquired knowledge of the trained model.

To generate a global explanation for deep neural networks trained with an unstructured dataset, we propose a feature-based rule explanation (FEB-RE) method to visualize high-level features and provide a logical explanation for humans. We first determine the correlation between the input units and high-level feature units and then, based on that, generate a visualized activation map for the feature units. Next, we generate explanations via decompositional approaches to high-level feature units with generate an if-then rule, where the premise is visualized feature units.

In addition, we identify the adverse effect of the complexity of the network structure on the interpretability of the explanation. We also introduce neural network pruning techniques with a simple heuristic to remove duplicated and complementary units to minimize adverse effects caused by the structural complexity.

The experimental results demonstrate that FEB-RE generates explanations that are not significantly different from the original neural network. We also prove that the heuristic method used for network pruning effectively reduces the network size and makes the generated explanation concise and comprehensible. To the best of our knowledge, this paper is the first attempt to generate a global explanation based on high-level features of the neural network.

Our main contributions can be summarized as follows: (i) We introduce a methodology for visualizing high-level features of trained fully connected neural networks. (ii) We present a global neural network interpretation technique based on high-level features. (iii) We propose a simple heuristic pruning technique that can remove duplicated and complementary units of neural networks.

The remainder of the paper is organized as follows. In Section 2, we review the XAI techniques for describing neural networks. Section 3 introduces the techniques used in the FEB-RE method in detail. Next, Section 4 describes the experiment settings and evaluation metrics, and Section 5 explains the experimental results demonstrating the FEB-RE. Finally, we conclude the paper and discuss future work in Section 6.

## 2. Related Works

### 2.1. Rule Extraction from Neural Networks

The study of extracting rules from neural networks trained with structured data is mainly classified into three approaches: Pedagogical, decompositional, and eclectic [4,10].

Pedagogical approaches extract rules from the neural network by analyzing the relationship or building interpretable models as a surrogate model with the input and output values of the neural network. These methods are often called black-box methods because they extract rules without knowing the internal structure of the neural network. For example, binarized input-output rule extraction [11] builds a truth table from the neural network input and output values and generates a ruleset. Validity-interval analysis [12] extracts the rule by assigning a random input value to the neural network and finding a stable section where the output value does not change while slightly changing the input value. Pedagogical approaches have the advantage of extracting rules at a low cost and using various interpretable models. However, these have been criticized for not interpreting neural networks but indirectly analyzing them through alternative models.

Conversely, decompositional approaches extract rules from every neural network unit, concatenating them to generate rules for the entire neural network model; these are also known as white-box approaches. KnowledgeTron (KT) [13] determines the set of connections activated by each perceptron through a breadth-first search and converts the set as an if-then rule in a disjunctive normal form, generating a ruleset for the entire neural network by rewriting rules from the output layer to the input layer. The ordered-attribute search (OAS) [14] is similar to the KT, but this algorithm extracts rules after sorting the

weights of the perceptron in descending order; thus, it is possible to prune unnecessary rules and quickly search the necessary rules. The continuous/discrete rule extractor via decision tree induction (CRED) [15] converts each perceptron into a decision tree based on the C4.5 algorithm with the training data then extracts the if-then rule from the tree. The biggest strength of decompositional approaches is that they generate rules through the process equivalent to how neural networks work. However, these approaches are challenging to use with neural networks with deep layers due to their shortcomings.

First, decompositional approaches place limitations on the activation function because asymmetry activation functions, such as the rectified linear unit (ReLU) function, are hard to convert into a binary rule or fuzzy logic since the range is different for each perceptron. In addition, the rule uncertainty, which is an error generated while converting the perceptron into a rule, increases as the layer deepens, resulting in a significant error in the rule for the entire neural network [16]. Most of all, the cost of decompositional approaches is the biggest problem since it is exponential [10]. Therefore applying the decompositional approach to the deep neural network [16–18] exhibits results with limited layers and unit size, and even under these conditions, failure often occurs due to the cost.

The eclectic approach is difficult to define clearly, but it takes advantage of the pedagogical and decompositional approaches. For example, the fast extraction of rules from neural networks (FERNN) [19] algorithm samples some of the training data and analyzes the activation value of the hidden unit based on sampled values, creating a C4.5 decision tree to generate the rules. The RX [20] creates a cluster based on the value of the hidden unit and creates a rule between the cluster and output. Then, it searches the input unit that activates the hidden unit and merges the two rulesets.

### 2.2. Neural Networks Explanation

Various techniques have been introduced to explain deep neural networks trained on unstructured image datasets [5,6,21]. Since it is difficult to cover all studies, we are briefly cover significant studies.

Layer-wise relevance propagation [22] is a technique to find units with high relevance that affects the classification result for each layer when a data instance is given and finally provides a visualized explanation by finding the relevance of the input units and classification result. CAM [7] represents where the neural network focuses by generating a heatmap based on the influence of each unit of the fully connected layer on classification with a given image instance. LIME [8] generates an explanation based on the classification result of random sampling instances adjacent to a given single instance. Xu, Kelvin, et al. presented the first attention-based technique that shows the features that are noticed in the process of classifying a given instance [9].

These are explanation techniques for neural networks that have trained images datasets but only provide local explanations. Since local explanation only explains for a specific instance, it cannot properly explain other instances. In addition, the local explanation only describes the classification process after input a given data instance to the neural network, not the knowledge learned by the neural network. Therefore a global explanation method to image dataset is necessary to figure out what knowledge the neural network has trained from the data domain, not the classification process of a given instance.

## 3. Feature-Based Rule Explanation

This paper proposes a FEB-RE method to explain the image dataset using an eclectic approach. The overall operation process is presented in Figure 1. The presented method consists of the following steps:

1. Training a fully connected neural network using datasets.
2. Defining one of the hidden layers to analyze the high-level features as a high-level feature layer. We assume that each hidden unit of the high-level feature layer has learned one high-level feature. After that obtaining the input units (low-level features) that activate the high-level feature unit.

3. Applying network pruning to the high-level feature layer. If there are unnecessary units in the layer, it becomes an obstacle to generating a concise explanation of the network and also increases the computation cost.

4. Obtaining a ruleset to activate the output layer using the pruned high-level features based on the decompositional approach. The generated explanation reveals how the entire neural network works by if-then rulesets through visualized high-level features.

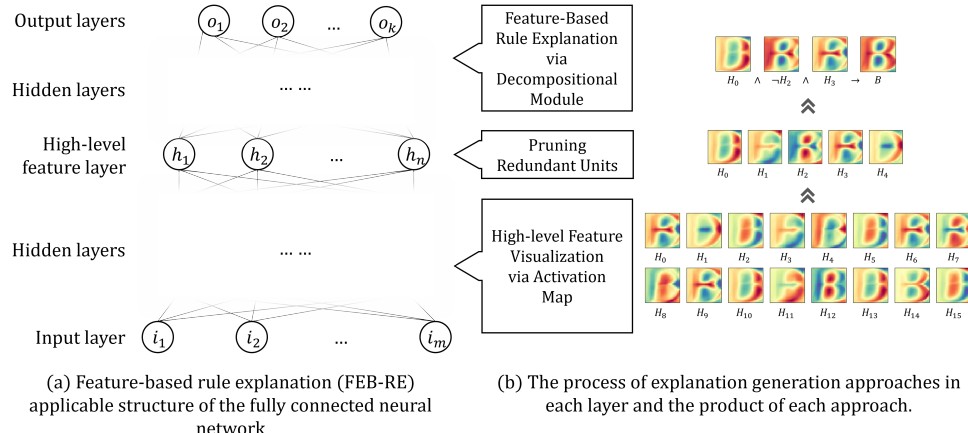

(a) Feature-based rule explanation (FEB-RE) applicable structure of the fully connected neural network

(b) The process of explanation generation approaches in each layer and the product of each approach.

**Figure 1.** Feature-based rule explanation (FEB-RE) applicable structure of the fully connected neural net-work and overall explanation generated approaches used in each layer.

The FEB-RE method presented in this paper has several advantages over the existing neural network explanation methods.

First, we only give a slight limitation to the activation functions. Since rules must be generated in all neural network units, the existing decompositional approaches, such as [13,14,16,23], must limit the activation function, making it difficult to cope with the vanishing gradient problem [24]. However, FEB-RE restricts a sigmoid function only for the high-level feature layer, thus other layers can choose activation functions freely. Therefore, proposed method is less affected by the vanishing gradient problem. In addition, because FEB-RE performs decompositional analysis only on limited layers, it can also handle problems caused by the uncertainty of the rules.

Second, FEB-RE can generate an explanation for the neural network with a feasible computational cost. FEB-RE does not apply a decompositional approach to all layers but considers the network size and arbitrarily adjust layers to be analyzed decompositionally, so cost constraints are less affected. Moreover, applying network pruning to subtract the redundant units minimizes the computational cost that may unnecessarily occur.

Third, FEB-RE does not interpret the neural network using the input unit directly; instead, it interprets it based on the high-level features learned by the neural network. When a neural network is analyzed using pedagogical approaches for unstructured datasets, the generated explanation is too complex for humans to understand, or a specific feature takes a large part in the explanation. Algorithms, such as CAM [7] or LIME [8], do not explain what the neural network has learned but only show where it focuses while classifying datasets. On the other hand, FEB-RE first analyzes and visualizes the high-level features of the trained neural network at the input unit level then generates an explanation for the entire network based on high-level features; therefore, FEB-RE can provide explanation for image data with logical form by visualizing high-level features.

### 3.1. Neural Network Structure

The FEB-RE can apply can be applied to the fully connected neural network with a classification task to generate an explanation. Training Dataset $D$ is defined as $D = \{(\mathbf{x}_1, \mathbf{y}_1), (\mathbf{x}_2, \mathbf{y}_2), \cdots, (\mathbf{x}_l, \mathbf{y}_l)\}$, where $\mathbf{x}$ is input instance, and $\mathbf{y}$ is output instance. Input in-

stance **x** is m-dimensional vectors with normalization, so $\mathbf{x} \in N^m$ and $N = \{x \mid 0 \le x \le 1\}$. output instance **y** is one-hot vector with k dimension, so $\mathbf{y} \in C^k$ and $C = \{0, 1\}$.

The structure of the fully connected neural network is depicted in Figure 1. Neural network consists of an input layer $I$, hidden layers, and an output layer $O$ like a general fully connected neural network. One of the hidden layers that we want to analyze is designated as a high-level feature layer $H$. Since input layer $I$ is composed of $m$ units, $I = \{i_1, i_2, ..., i_m\}$, where $i$ means input unit. Similarly, if there are $n$ feature units $h$ in high-level feature layer, $H$ can be defined as $H = \{h_1, h_2, ..., h_n\}$. The output layer $O$ should be composed of $k$ units as same as the output data, so $O = \{o_1, o_2, ..., o_k\}$, where $o$ means output unit.

Any activation functions, such as ReLU, leaky ReLU, or ELU, can be applied to other hidden layers, but the high-level feature layer is limited to only using the sigmoid activation functions. Restricting the activation function of the high-level feature layer limits the range of values while visualizing the feature activation map and generating explanations by the decompositional method.

To minimize the uncertainty of the rule that occurs in the neural network interpretation process, we apply hidden unit clarification [16] as a regularization term with the high-level feature layer and hidden layers higher than the high-level feature layer as follows:

$$Loss_h = Loss + c \sum_j min\{1 - h_j, h_j\}. \tag{1}$$

With hidden unit clarification, we can train the neural network weights so that the activation values approximate the maximum or minimum values of the activation function.

### 3.2. High-Level Feature Visualization via an Activation Map

We represent high-level features by visualizing input unit combinations that activate feature units of high-level feature layers, making it possible to overcome the limitation of previous studies that did not indicate knowledge the neural network acquired. To determine the input unit effects on the activation of the feature unit, we used a modification of the Pearson correlation coefficient [25]. The range of the input units and feature units is limited through normalization of the input data and the limitation of activation function; thus, there is no problem using the Pearson correlation coefficient.

The correlation between the single high-level feature $h$ and the single input unit $i$, $\rho_{h,i}$, can be expressed as follow:

$$\rho_{h,i} = \frac{\sum_j^l (h(\mathbf{x}_j) - \overline{h(\mathbf{x})})(i(\mathbf{x}_j) - \overline{i(\mathbf{x})})}{\sqrt{\sum_j^l (h(\mathbf{x}_j) - \overline{h(\mathbf{x})})}\sqrt{\sum_j^l (i(\mathbf{x}_j) - \overline{i(\mathbf{x})})}}, \tag{2}$$

where $h(\mathbf{x}_j)$ means the activation value of the high-level feature when the $j^{th}$ training instance comes in, $i(\mathbf{x}_j)$ means the value corresponding to the input unit among the training instances (i.e., the $j^{th}$ value of vector **x**). In addition, $\overline{h(\mathbf{x})} = \sum_j^l h(\mathbf{x}_j)/l$, and $\overline{i(\mathbf{x})} = \sum_j^l i(\mathbf{x}_j)/l$, each represent the expected values of $h$ and $i$ with dataset $D$. According to the Cauchy-Schwarz inequality, $\rho_{h,i}$ has a value between $+1$ and $-1$. As the value of $\rho_{h,i}$ is closer to $+1$, it means that input unit $i$ has a effect on activating $h$, and conversely, as the value of $\rho_{h,i}$ is closer to $-1$, it means that $i$ has an effect on deactivating $h$.

However, we need to extend the above formula since we need to find all input units that activate the feature unit. Therefore, we need to find the correlation $\rho_{h,I}$ between the single feature unit $h$ and all input units $I$ for all feature units in $H$ as obtained in Equations (3) and (4):

$$\rho_{h,I} = \{\rho_{h,i_1}, \rho_{h,i_2}, \cdots, \rho_{h,i_m}\}, \tag{3}$$

$$\rho_{H,I} = \{\rho_{h_1,I}, \rho_{h_2,I}, \cdots, \rho_{h_n,I}\}. \tag{4}$$

Figure 2 is a visualized activation map of the features obtained by correlating hidden units and input units. If input units are involved in activating the hidden unit, the units are visualized as red. Conversely, if the input unit deactivates the hidden unit, it is visualized as blue, and if there is no correlation, the unit is visualized as yellow. The proposed method enables observing the shape of each high-level feature, not just the focus area of the neural network.

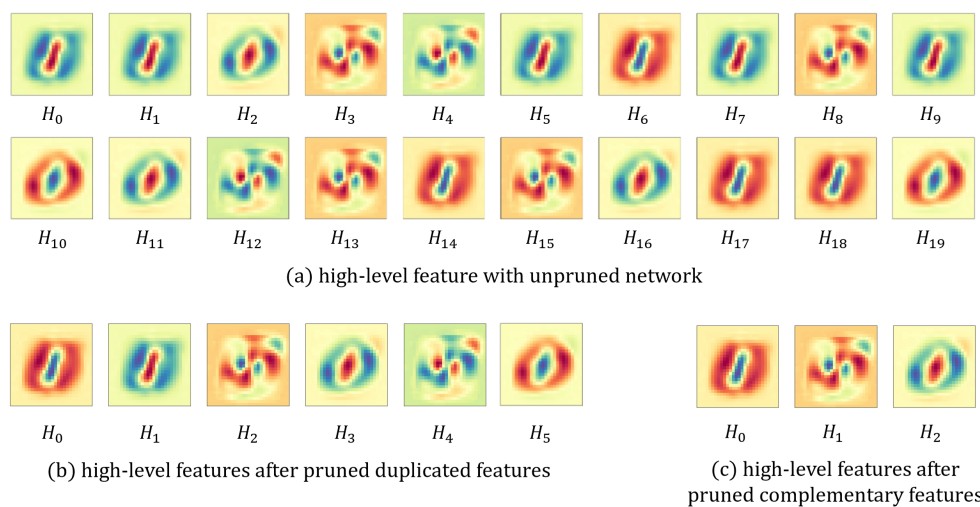

(a) high-level feature with unpruned network

(b) high-level features after pruned duplicated features

(c) high-level features after pruned complementary features

**Figure 2.** Visualized activation map of feature units and the pruning process of a neural network trained on MNIST data. Visualized feature units are red if the input units influence the feature to be active; otherwise, they are blue. (**a**) Visualized high-level features of an unpruned neural network. (**b**) Pruning duplicated high-level features in (**a**), but with some features complementary to each other. After removing complementary features, the most optimized features are in (**c**).

### 3.3. Pruning Redundant Units

Selecting the optimal hidden unit size is one of the issues in using neural networks. Recently, with the development of technology, rather than finding the number of hidden units suitable for the data domain, the size of the hidden unit is set generously. While to train a neural network with lower error, unnecessary hidden units are not a substantial problem, but such unnecessary units could cause difficulty during interpreting neural networks.

Studies like [20,26] have used the technique to prune the inactivated unit to handle these problems. However, we applied two other network pruning techniques using simple heuristics: Duplicated unit pruning and complementary unit pruning. We did not prune inactivated units because both OAS and CRED methods, which we used as decompositional modules, are hardly affected by less influential units, and inactivated units are merged into a single unit while pruning duplicated units.

First, to remove duplicated units activated by the same or similar input units among the feature units, we calculated the similarity between hidden units to find which units were duplicated. The similarity between the two hidden units $h_a$ and $h_b$ for the training data $D$ can be obtained using Equation (5):

$$S(a, b; D, h) = 1 - \frac{\sum_i^l \|h_a(\mathbf{x}_i) - h_b(\mathbf{x}_i)\|}{l}. \tag{5}$$

For each data input $\mathbf{x}$ of the dataset $D$, the similarity of the hidden unit activation values was obtained by calculating the euclidean distance $\|h_a(\mathbf{x}_i) - h_b(\mathbf{x}_i)\|$ between the two hidden unit activation values. High-level feature layer use sigmoid function as activation function, so $0 < h(\mathbf{x}) < 1$, and therefore the condition $0 < \|h_a(\mathbf{x}_i) - h_b(\mathbf{x}_i)\| < 1$ is satisfied. However, the distance between $h_a$ and $h_b$ is close to 0 as the difference of activation values is small, while the similarity $S$ should have a value close to 0 when the

difference is big. Therefore, similarity can be obtained by simply subtracting distance from 1, and also satisfies $0 < S < 1$.

Duplicated units were selected from feature units that exceed the specified threshold value $\tau_r$ after calculating the similarity between all feature units. After obtaining groups of feature units whose similarity exceeds $\tau_r$, the input weights of the new feature units were assigned by averaging the input weights of the feature units belonging to the same duplicated group. Similarly, after summing all the connection weight values to the next layer and assigning them to a new unit, duplicated units were removed, and a new unit was connected to the neural network.

During the neural network training process, we found a tendency to learn not only duplicated but also complementary units, which are activated with the same inputs while the result of activation is the opposite. These complementary units also must be removed because overlapping features appear when analyzing the neural network, making interpretation difficult.

To overcome this problem, we applied an additional pruning technique to remove complementary units. The approach is similar to that used when pruning duplicated units. Figure 3 illustrates the heatmap of similarity and visualized feature units after removing duplicated feature units. The heatmap reveals that $H_2$ and $H_{15}$, $H_3$ and $H_6$, and $H_4$ and $H_{17}$ have low similarity. In these feature units, the input units have opposite effects on activation. Therefore, it is reasonable to prune complementary units based on similarity.

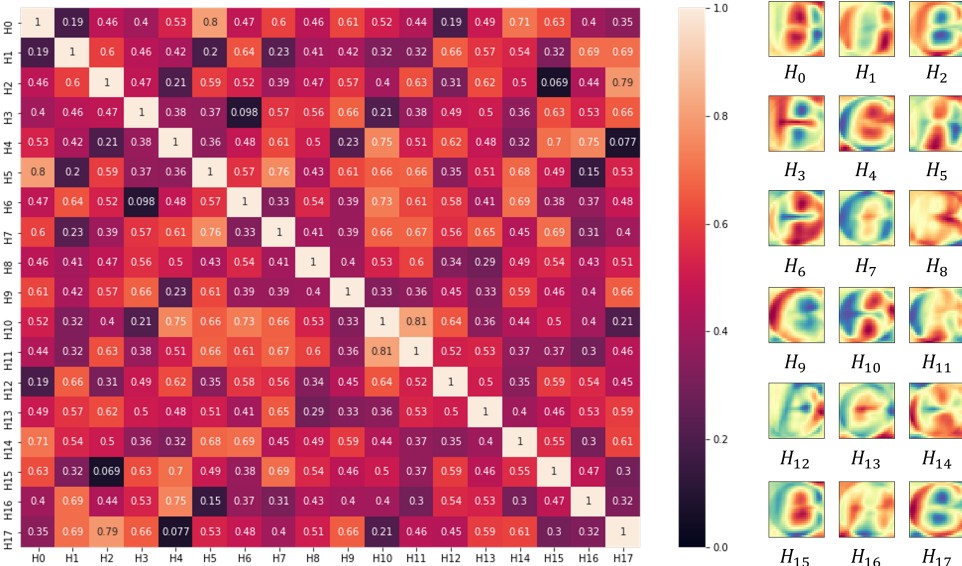

**Figure 3.** Heatmap of the similarity between feature units and visualized high-level features after pruning duplicated units in the neural network trained with the notMNIST dataset. Feature units with low similarity have a complementary relationship.

To prune the complementary units, we calculate the similarity between the feature units and define the feature units whose similarity does not exceed $\tau_s$ as a complementary relation. After that, pruning starts with the complementary units with the lowest similarity. We do not average the input weights this time but only use input weights from one unit because we found that the weights are not trained to be opposite even in the complementary relationship—still, the weight values to the next layer summed after reversing the weights of one unit.

Figure 2 presents the overall process of neural network pruning. This example reveals the process of changing the high-level features of the neural network trained to classify 0, 1, and 5 on the MNIST dataset through pruning. The visualized high-level features of the trained neural network with 20 high-level feature units are presented in Figure 2a, where $H_0$, $H_1$, $H_5$, $H_7$, $H_9$ are duplicated feature units, and $H_3$, $H_8$, $H_{14}$, and $H_{16}$ are

another duplicated feature units. The results in Figure 2b are derived after the duplicated network pruning process. When comparing Figure 2a,b, most overlapping feature units are merged into one feature, leaving only six feature units. Moreover, $H_0$ and $H_5$ or $H_1$ and $H_3$ appear to be similar features, but there is a difference in the area correlated with the low-level feature, and the area with a higher correlation value also has a difference. However, high-level features $H_0$ and $H_1$, $H_2$ and $H_4$, and $H_3$ and $H_5$ are still complementary. Finally, the result of removing these complementary feature units is in Figure 2c, leaving only three units.

### 3.4. Feature-Based Rule Explanation via the Decompositional Module

We used decompositional methods as an explanation-generating module to generate explanations for the entire neural network using high-level feature units. Any rule extraction method introduced in Section 2.1 can be used as an explanation-generating module. However, because the rule generation process that concatenates the analyzed results of each unit is similar to the nature of a neural network, we used decompositional methods only.

Consequentially, we used the OAS method [14], which searches for rule combinations that activate the perceptron, and the CRED method [15], which generates a rule with a C4.5 tree based on the data. The KT method [13] and Tsukimoto's algorithm [23] were not used because they are identical to OAS, and the M-of-N method [27] was excluded because this approach was not suitable for pruned neural network models.

### 4. Experiments

### 4.1. Datasets

To demonstrate the performance of FEB-RE, we experimented based on image datasets. As FEB-RE can only be applied to fully connected neural networks, there is a limitation in the image dataset that can be used due to the neural network performance. We used the numerical image dataset MNIST [28], the alphabet image dataset notMNIST [29], and the clothing image dataset Fashion-MNIST [30] for the experiments.

The image process for a fully connected neural network is different from that of humans; thus, results may not be clear for humans with full datasets. Therefore, we prepare additional partial datasets to demonstrate that interpretable explanations with high-level features can be generated with FEB-RE. The full dataset uses the entire dataset, and the partial dataset represents a dataset extracted by selecting four categories that present feature differences of the whole dataset. We selected 0, 1, 5, and 8 for NMIST; B, D, F, and I for notMNIST; and top, pants, shoes, and bag for Fashion-MNIST. The details of the three datasets used in the experiment are in Table 1.

**Table 1.** Datasets for experiment.

| Dataset | Train Instance | Test Instance | Features | Categories |
|---|---|---|---|---|
| MNIST partial | 23,937 | 3981 | 784 (28 × 28) | 4 |
| MNIST Full | 60,000 | 10,000 | 784 (28 × 28) | 10 |
| notMNIST partial | 5992 | 1498 | 784 (28 × 28) | 4 |
| notMNIST Full | 18,724 | 3744 | 784 (28 × 28) | 10 |
| Fashion-MNIST partial | 24,000 | 4000 | 784 (28 × 28) | 4 |
| Fashion-MNIST Full | 60,000 | 10,000 | 784 (28 × 28) | 10 |

### 4.2. Implementation Details

We used the TensorFlow 2.4.1 library to implement and train the neural network. Fully connected neural networks with four hidden layers of the 784-400-200-100-50-10 structure were used for the full dataset. For the partial dataset, 784-200-100-50-16-4 fully connected neural networks with four hidden layers were used.

In FEB-RE, the high-level feature layer can be selected from any hidden layer, so we selected the highest hidden layer as the high-level feature layer in the experiment. All

hidden layers except the high-level feature layer used the ReLU activation function. The high-level feature layer used the sigmoid activation function, and the output layer used the softmax function for classification.

We used cross-entropy as a loss function to obtain the classification loss. Additionally, we used the hidden unit clarification [16] as a regularization term of the high-level feature layer while applying the clarification constant $c = 0.5$. We used the Adam optimizer [31] with the default parameters for the learning rate $l = 0.001$, $\beta_1 = 0.9$, $\beta_2 = 0.999$, and $decay = 0$. The batch size was set to 32 and trained for up to 20 epochs. At this time, we separated 10% of the training data as validation data and applied an early stopping technique based on validation loss.

Table 2 presents the training accuracy and test accuracy of each full dataset according to the changes in the pruning threshold parameter $\tau_r$. It shows that the number of high-level features and accuracy according to the $\tau_r$ have a trade-off relationship. While $\tau_r = 0.95$, there was little change in training and test accuracy, but it did not significantly reduce the network features. The number of high-level features remarkably decreased when $\tau_r = 0.9$. After that, there is only a slight change in the accuracy and number of high-level features, and then the accuracy decreases sharply when the $\tau_r = 0.75$. We set $\tau_r$ to 0.9 since we want to prune features with higher similarity while appropriately reducing the high-level feature unit. We set the complementary unit pruning threshold $\tau_s$ as 0.1, which is the symmetric value of $\tau_r$. After pruning the redundant units in the retraining process, we trained the neural network for only one epoch with the training dataset with a batch size of 32.

**Table 2.** Comparison of training and test accuracy of networks according to changes in threshold parameters $\tau_r$ used for pruning duplicated high-level features. Since there was no significant difference in accuracy while the threshold $\tau_r$ was between 0.8 and 0.9, we used 0.9 as $\tau_r$ to prune features with high similarity.

| Dataset | MNIST Full | | | notMNIST Full | | | Fashion-MINST Full | | |
|---|---|---|---|---|---|---|---|---|---|
| Threshold $\tau_r$ | # of features [1] | Training accuracy | Test accuracy | # of features | Training accuracy | Test accuracy | # of features | Training accuracy | Test accuracy |
| unpruned | 50 | 99.70 | 97.90 | 50 | 98.10 | 97.36 | 50 | 93.32 | 88.98 |
| 0.95 | 34 | 99.69 | 96.81 | 31 | 97.77 | 97.08 | 38 | 90.5 | 86.91 |
| 0.9 | 18 | 97.68 | 95.6 | 18 | 96.77 | 95.86 | 20 | 86.97 | 83.56 |
| 0.85 | 15 | 97.46 | 95.17 | 14 | 96.32 | 94.75 | 17 | 86.26 | 82.87 |
| 0.8 | 13 | 97.21 | 94.67 | 12 | 95.35 | 94.4 | 15 | 84.66 | 81.59 |
| 0.75 | 10 | 95.74 | 92.35 | 10 | 93.16 | 91.32 | 12 | 82.02 | 79.29 |

[1] of features denotes number of high-level features.

When using the OAS method, we set the threshold for activation to sigmoid 0.8 (0.2 for negation), so if the activation value of the combination was smaller than that, it was not treated as a rule. In addition, we extracted only rules with the same length as the premise with the first generated simplest rule because complex rules with many premises have poor interpretability.

The CRED method generates rules based on the C4.5 tree algorithm [32]. A tree was built based on the high-level feature unit activation value and the output unit classification categories when we put the training data into the neural network. Decision tree pruning based on the confidence factor was applied with a confidence level $CF = 0.25$ to prevent the tree from overfitting and the generated rule from being overly complicated.

### 4.3. Evaluation Metrics

To quantitatively prove the performance of the explanation generated by FEB-RE, we present the accuracy, fidelity, average coverage, and the number of the explanations based on [16,17].

Accuracy measures how precisely given datasets are classified while using the generated explanation as a classifier. We aimed to determine how well the method explains the

trained knowledge of the neural network, not whether it learns knowledge generously that has not been previously encountered, so we evaluated the accuracy with the training data, not the testing data.

Fidelity measures the differences in classification output between the original model and generated explanation, given the same input data. In other words, it measures how well the generated explanations approximate the original neural network. Fidelity is also measured using the training data.

Accuracy and fidelity are closely related indicators. If the neural network model has high accuracy and the generated explanation has high fidelity, the explanation has high accuracy. However, in some cases, such as generating a concise and general explanation from the neural network that is not generalized due to overfitting, a situation may occur where the accuracy of the explanation is high, but the fidelity is low.

Coverage and the number of explanations are evaluation metrics indicating the explanation quality. Coverage indicates the percentage of data that a single explanation can represent, and average coverage is an evaluation metric that divides the coverage of each explanation by the total number of explanations. High coverage means that one explanation can cover many data, and high average coverage and accuracy with a small number of explanations indicate that a small number of explanations can sufficiently cover most data.

However, because the original coverage metric was a formula calculated for the rule generated from models trained with a structured dataset, it can count which data are covered by the rule. In contrast, FEB-RE generates an explanation for unstructured image data, and we must modify the formula for calculating the existing coverage as follows:

$$coverage = \frac{\sum^{l} \frac{\sum^{m} \|\mathbf{x}_m - Exp(m)\|}{m}}{l},$$ (6)

where $Exp$ represents flattened image vector generate by explanation. Since the explanation of FEB-RE is expressed by the IF-THEN rule that assumes high-level features as a premise in DNF, it is possible to generate a representative image of conclusion. $Exp(m)$ is $m^{th}$ value of generated explanation and $\mathbf{x}_m$ means $m^{th}$ value of input vector $\mathbf{x}$. Therefore, the modified coverage indicates the average of all distances between the representative image vector of the explanation and the input data vector.

Additionally, to determine the performance of network pruning, we compared the number of feature units, training accuracy, testing accuracy, and number of explanations for each unpruned network, network-pruned duplicated units, network-pruned duplicated and complementary units, and the retrained network after pruning.

## 5. Results

### 5.1. Effects of Pruning Network

Table 3 lists the results for the number of feature units, training accuracy, testing accuracy, and number of explanations for each unpruned network, network-pruned duplicated units, network-pruned duplicated and complementary units, and the retrained network after pruning on the three full datasets. During the network pruning experiment, only OAS was used as the decompositional module for generating explanations.

**Table 3.** Comparison of network pruning performance on MNIST, notMNIST, and Fashion-MNIST full datasets. The pruned and retrained neural network has only a slight performance difference compared to the unpruned neural network. However, the number of explanations is significantly reduced.

| Dataset | Model | # of High-Level Features | Training Accuracy | Test Accuracy | # of Exps |
|---|---|---|---|---|---|
| MNIST Full | unpruned | 50 | 99.70 | 97.90 | - |
| | pruned duplicate | 18 | 97.68 | 95.60 | 335 |
| | pruned complement | **16** | 92.18 | 90.68 | 78 |
| | pruned+retrained | **16** | **99.72** | **97.75** | **70** |
| notMNIST Full | unpruned | 50 | 98.10 | 97.36 | - |
| | pruned duplicate | 18 | 96.77 | 95.86 | 737 |
| | pruned complement | **15** | 91.18 | 90.65 | 103 |
| | pruned+retrained | **15** | **97.65** | **96.26** | **75** |
| Fashion-MNIST Full | unpruned | 50 | 93.32 | 88.98 | - |
| | pruned duplicate | 20 | 86.97 | 83.56 | 3368 |
| | pruned complement | **16** | 80.91 | 79.14 | 223 |
| | pruned+retrained | **16** | **92.77** | **88.27** | **165** |

During pruning of the duplicated high-level features in all three datasets, the layer size was reduced by more than 60%, and the model performance loss is slight without a retraining process. However, removing the complementary high-level features causes a significant loss of performance. This phenomenon is presumed to occur because the weights cannot be trained oppositely, even if the high-level feature layer causes opposite activations. However, after retraining the pruned neural network with only one epoch, a negligible difference exists in the training and testing accuracy between the unpruned network and the final result.

While no significant change exists in the accuracy, the number of explanations reduced significantly after the network pruning process. The number of high-level features in the unpruned network is enormous, so the number of explanations could not be obtained with the limited memory and time. As pruning progresses, the number of generated explanations is reduced by 4 to 17 times because the combination of weights in the perceptron is exponential. The retraining process exhibits a slight decrease in the number of explanations, despite no change in the unit size, which reduces the combination produced by training the values of the weights more polarized to 0 or 1.

In Figure 4, we take the notMNIST partial dataset as an example to display the high-level features and explanations that change according to the pruning progress. Figure 4a illustrates the high-level features of the unpruned network, and there are many identical high-level features. For example, $H_0$, $H_6$, and $H_9$ are similar, as are $H_2$, $H_5$, and $H_{15}$. Figure 4b presents one of the explanations generated in the unpruned network and constitutes one explanation with 10 premise terms. Similar to high-level features, overlapping premise terms exist, like $H_6$, $\neg H_7$, and $H_9$ in the explanation, and such terms make an explanation complicated to interpret.

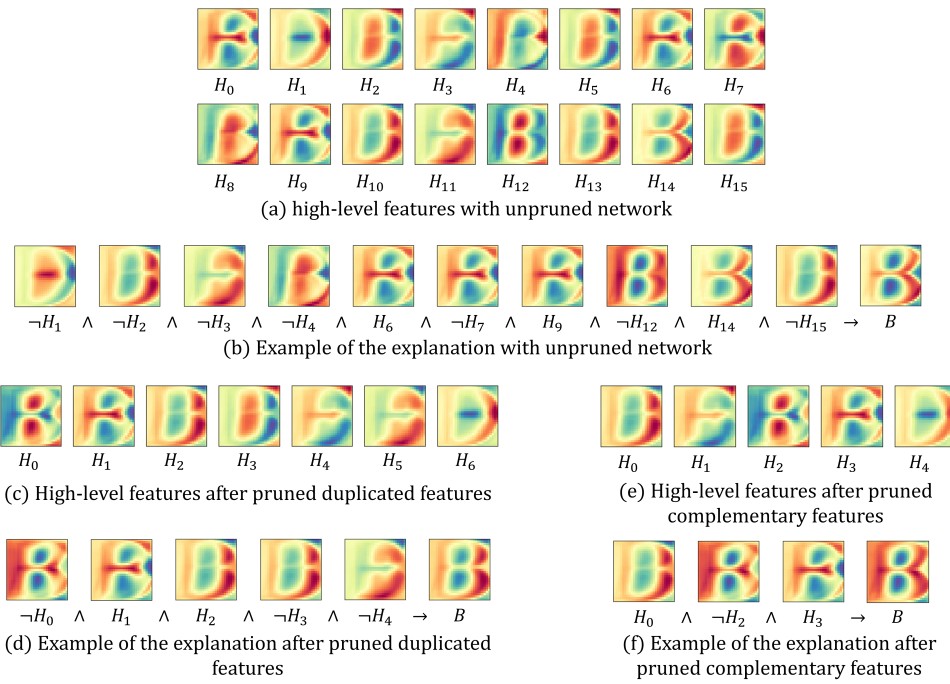

**Figure 4.** (**a**) Feature units of an unpruned neural network. (**b**) Example of an explanation generated using unpruned features, which have many duplicated features. (**c**) Pruning results for duplicated feature units. (**d**) Explanation generated from the feature in (**c**), which still has duplicate premises due to complementary features. (**e**) Pruning results for complementary features where five features remain. (**f**) Final generated explanation.

Figure 4c visualizes the high-level features generated in the network that removed duplicated features, and the number of high-level features reduced from 16 to seven. However, some features are in a complementary relationship, such as $H_2$ and $H_3$, or $H_4$ and $H_5$. Figure 4d is an example of the network explanation with removed duplicated features, and the length of the premise term is reduced and simplified to five. However, overlapping terms, such as $H_2$ and $\neg H_3$, still exist due to the complementary term.

Finally, the number of high-level features is reduced to five in Figure 4e, which pruned the complementary features. Figure 4f reveals that the final generated explanation has only three premise terms and high interpretability without overlapping terms.

*5.2. Quantitative Analysis of Explanation*

Table 4 compares the accuracy, fidelity, average coverage, and number of explanations for all datasets. Original indicates the performance of the neural network model, which is the base model for comparison with the generated explanation. After the pruning and retraining processes, we used OAS and CRED as the decompositional modules to generate rules from the feature units and generated explanations.

In most cases, generated explanations based on the CRED module perform best in all evaluation metrics, with even higher accuracy than the original neural network. Unlike CRED, experiments with the OAS module have slightly lower accuracy than the original neural network in all cases. While the accuracy decreased, explanations with the OAS module also provide a sufficiently high-quality explanation. The accuracy of the generated explanations with CRED module is higher than that of the original neural network because it produced the generalized explanations.

**Table 4.** Comparison of accuracy, fidelity, coverage, and number of explanations generated by proposed feature-based rule explanation(FEB-RE) with the OAS and CRED modules for all datasets. The FEB-RE with CRED module generates high accuracy and high-quality explanations with a small number of explanation.

| Dataset | Model | Accuracy | Fidelity | Average Coverage | # of Exps |
|---|---|---|---|---|---|
| MNIST Partial | original NN | 99.96 | - | - | - |
| | FEB-RE+OAS | 99.65 | 99.67 | 23.13 | 29 |
| | FEB-RE+CRED | **99.96** | **99.97** | **31.97** | **16** |
| MNIST Full | original NN | 99.70 | - | - | - |
| | FEB-RE+OAS | 98.20 | 98.19 | 8.08 | 70 |
| | FEB-RE+CRED | **99.77** | **99.78** | **10.79** | **38** |
| notMNIST Partial | original NN | 99.24 | - | - | - |
| | FEB-RE+OAS | 98.71 | 98.89 | 24.43 | 28 |
| | FEB-RE+CRED | **99.43** | **99.23** | **25.35** | **14** |
| notMNIST Full | original NN | 98.10 | - | - | - |
| | FEB-RE+OAS | 97.65 | **97.89** | **9.19** | 75 |
| | FEB-RE+CRED | **99.25** | 97.14 | 8.03 | **42** |
| Fashion-MNIST Partial | original NN | **99.77** | - | - | - |
| | FEB-RE+OAS | 97.59 | 97.60 | 22.98 | 26 |
| | FEB-RE+CRED | 99.69 | **99.98** | **24.97** | **14** |
| Fashion-MNIST Full | original NN | 93.32 | - | - | - |
| | FEB-RE+OAS | 92.77 | 92.92 | 8.86 | 165 |
| | FEB-RE+CRED | **94.54** | **95.06** | **11.45** | **52** |

In Section 5.4, we analyze the cause of the difference in performance when using the CRED and OAS modules in more detail.

### 5.3. Qualitative Analysis of Explanation

We designed the following experiment to compare FEB-RE with other XAI techniques. We generated explanations with CAM [7], LIME [8], and LRP [22] techniques using MNIST, notMNIST, and fashion-MNIST partial datasets. We only provide qualitative analysis of these approaches with FEB-RE because it is difficult to compare quantitatively due to the difference of the explanation representation.

LIME generated explanations from the same fully connected neural network model that FEB-RE used and applied Quickshift image segmentation [33] while generating explanation. CAM and LRP generated explanations from LeNet [34], and CAM especially apply global average pooling.

Examples of the explanation generated by each technique are presented in Figure 5. In Figure 5, the highlighted area with red means positive effect on classification, the area with blue means negative effect, and the area with yellow does not affect the classification result.

Figure 5a,c,e show explanations generated by CAM, LIME and LRP. They provide explanations for given instances only since these are techniques for generating local explanations. CAM and LRP approaches represent the area or the shape where the neural network is focusing. The LIME visualizes the segmentation that affects the classification after creating a segment from the image. These explanations give us good inspiration, but since these techniques only tell where the neural network is focusing, reasoning by humans is essential to understand why the neural network made this decision. Looking at the explanation for number 8 in Figure 5a for example, we can only guess that the crossed shape has a significant influence on determining the number 8 because the heatmap generated by the CAM is concentrated in the center.

Figure 5b,d,f represent samples of the explanations generated by FEB-RE. Since FEB-RE is a global XAI technique, it can generate an explanation without any given instance and also provide a general explanation for each class, unlike local XAI approaches. Moreover,

while the local explanation can be applied to a specific instance, the global explanation provided by FEB-RE does express the general decision-making process of the neural network model. In addition, FEB-RE not only represents the heatmap of the final result but also provides explanations in the logical form with high-level features as premises, so it is possible to understand how each feature affects the final decision.

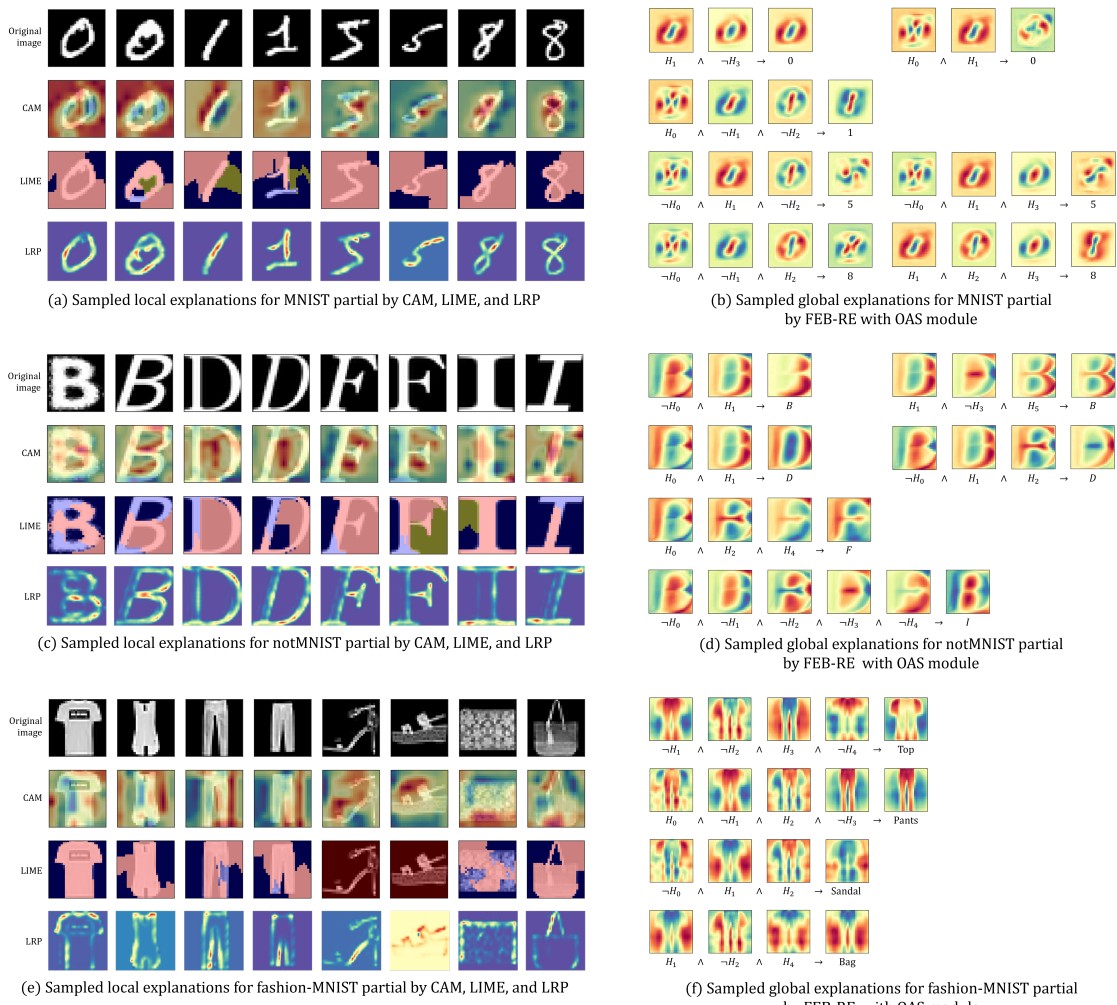

(a) Sampled local explanations for MNIST partial by CAM, LIME, and LRP

(b) Sampled global explanations for MNIST partial by FEB-RE with OAS module

(c) Sampled local explanations for notMNIST partial by CAM, LIME, and LRP

(d) Sampled global explanations for notMNIST partial by FEB-RE with OAS module

(e) Sampled local explanations for fashion-MNIST partial by CAM, LIME, and LRP

(f) Sampled global explanations for fashion-MNIST partial by FEB-RE with OAS module

**Figure 5.** Example of explanations generated by CAM, LIME, LRP and FEB-RE with the MNIST, notMNIST and fashion-MNIST datasets. The highlighted area with red means positive effect on classification, the area with blue means negative effect, and the area with yellow does not affect the classification result. CAM, LIME, and LRP provide an explanation for a specific instance since they are local explanation techniques. On the other hand, FEB-RE generates general global explanations in logical form.

### 5.4. Comparison of OAS and CRED

Figure 6 compares feature-based rule explanation(FEB-RE) using the OAS and CRED modules from the same neural network that trained the Fashion-MNIST partial dataset. Figure 6a presents the visualized result of the high-level features of the neural network. Among the explanations generated by each module, we selected those that share the same features in each category. The results in Figure 6b,c indicate that explanations with the OAS module have three to four premise terms, whereas the CRED module has one to two premise terms. Comparing the representative activation map of the conclusion generated by explanation reveals slight differences.

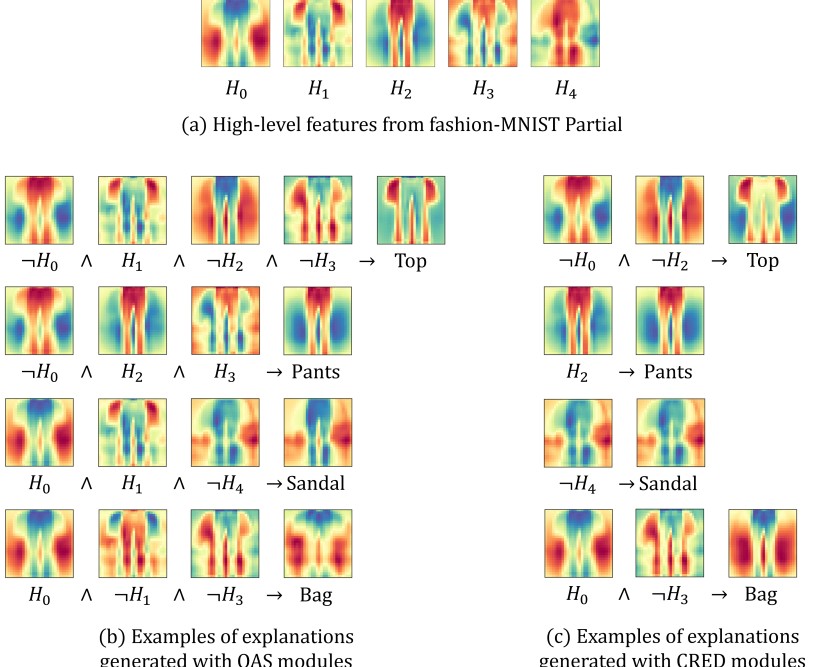

(a) High-level features from fashion-MNIST Partial

(b) Examples of explanations
generated with OAS modules

(c) Examples of explanations
generated with CRED modules

**Figure 6.** Comparison of explanations generated by feature-based rule explanation(FEB-RE) with the OAS and CRED modules with the Fashion-MNIST partial dataset. Explanations generated by the OAS and CRED modules present similar representative activation maps. However, the CRED module has a similar effect with shorter premise terms.

Taking a top as an example, both arms significantly influence activation, but there is a difference in the body part. The representative activation map of the OAS module indicates that the body part has some influence on deactivation, but in the CRED module, the body part has little effect on activation. Considering human common sense, it is nonsense to have a negative effect on classifying a body part for a top, so the explanation generated by the CRED module can be considered concise and correct. In other words, the rule generated by the OAS module uses two more unnecessary premise terms and generates an inaccurate explanation from the human viewpoint.

It is easier to understand the experimental results in Table 4 by considering the results of Figure 6. The reason the CRED module has high accuracy is that it generates a simple and reasonable explanation. Because the explanation generated by the OAS module is more complicated than the CRED module by adding a premise, even if it is the same explanation, the number of explanations also increases. Furthermore, the average coverage is also higher because the expansion created by the CRED module is more general than that with the OAS module. Based on the experimental results in Table 4 and Figure 6, it seems that the CRED module is superior to the OAS module in all aspects.

Figure 7 presents randomly generated sample images for each category from explanations with the OAS and CRED modules. The sample image is different due to the representative activation map in the process of generating it. The representative activation map was generated using only feature units that appeared in the premises, whereas the sample image considers the influence of feature units that did not appear in the premises. In the if-then rule, terms that do not appear in the premise are "don't care" terms, so whether the term exists or not should not affect the conclusion. Therefore, we created the sample image by randomly assigning weights between $-1$ and 1 to terms that do not exist in the explanation.

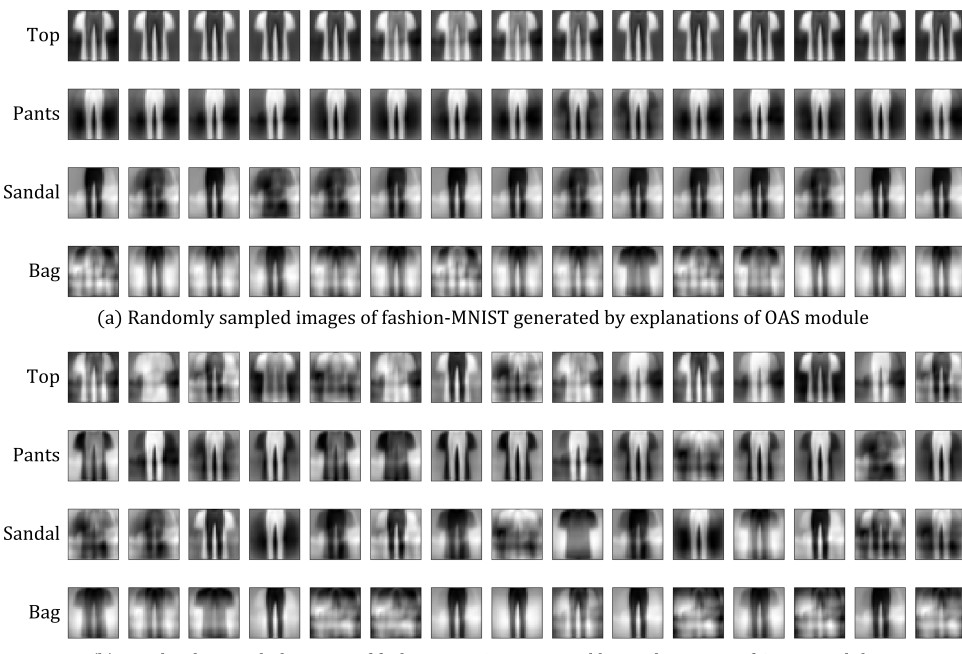

(a) Randomly sampled images of fashion-MNIST generated by explanations of OAS module

(b) Randomly sampled images of fashion-MNIST generated by explanations of CRED module

**Figure 7.** Randomly sampled image of the Fashion-MNIST partial dataset generated from the explanation of the feature-based rule explanation (FEB-RE) with the OAS and CRED modules for each category. The generated image in the explanation of the OAS module is stable, whereas the CRED module has a significant difference between images and has incorrect images.

Significant differences exist between random sample images in the expansion of the two modules. The sample images generated by the CRED module display significant changes, whereas the sample images generated by the explanation of the OAS module maintain a stable form. In Figure 7b, which is images generated by the explanation of the CRED module, the 10th, 12th, and 14th images of a top seem to be closer to pants than a top.

These results are based on the nature of the OAS and CRED modules. The OAS module generates the explanation from the weight combination of the perceptron's input that activates the unit, whereas the CRED module generates the explanation based on the input values and activation values of the unit. In particular, since the CRED module uses only the training dataset as an input, the generated explanation is overfitted to the training dataset. In Figure 5b,c, the CRED module generates an explanation for pants using only one premise, $H_2$. On the other hand, the OAS module adds two additional premises $H_0$ and $H_3$, which seem unnecessary. The reason that there are no $H_0$ and $H_3$ in the premise of the explanation generated by the CRED module is that there were no data affected by $H_0$ and $H_3$ in the training dataset for pants, so it was not necessary to consider these premises. On the other hand, since the OAS module generates an explanation that activates the perceptron based on the weight, the hidden influence of high-level features can be considered regardless of the input value.

To verify the stability of the description generated by each module, we generated 200,000 randomly sampled images from explanations of each decompositional module then obtained the fidelity with the original neural network results. The experimental results shown in Table 5 indicated that the explanations of the OAS module had higher fidelity for all six datasets. However, the generated explanations of the CRED module reached different conclusions from the original network more than half of the time in the worst cases, such as the MNIST full and Fashion-MNIST full datasets.

Because explanations generated by the OAS and CRED modules have pros and cons, it is difficult to determine which module is better. An analysis using the OAS module is the right choice to accurately understand the neural network decision-making process. If the range of the data domain is not much different from the training data, it would be better to use the CRED module to provide a more comprehensible explanation.

**Table 5.** Comparison of fidelity between randomly sample images from feature-based rule explanation(FEB-RE) of each decompositional module and original neural network. For all datasets, the explanation generated from the OAS module has higher fidelity.

| Model | Dataset | | | | | |
|---|---|---|---|---|---|---|
| | MNIST Full | MNIST Partial | notMNIST Full | notMNIST Partial | Fashion-MNIST Full | Fashion-MNIST Partial |
| FEB-RE+OAS | **91.46** | **95.84** | **95.11** | **98.20** | **84.97** | **92.30** |
| FEB-RE+CRED | 48.79 | 69.64 | 64.59 | 73.99 | 43.72 | 67.86 |

## 6. Conclusions

We propose the FEB-RE method to generate an explanation for the fully connected neural network trained with image dataset. First, we developed the activation map based on the correlation between the input and feature units to visualize the high-level feature units of the trained neural network in the human-recognizable form. Second, we generated the if-then rule with feature units and explained the knowledge of the trained neural network in the interpretable form. Third, we introduced a pruning technique to remove duplicated and complementary units to provide a more concise and comprehensible explanation.

Experiments showed that the introduced simple heuristic pruning technique effectively removes unnecessary high-level features while having little effect on the performance of the neural network. Moreover, we prove that the generated explanation by the FEB-RE method can sufficiently cover the knowledge of the original trained neural network. In addition, we show the advantages of the proposed method through a qualitative comparison with the previously studied XAI techniques. Moreover, we used two different methods to create explanations and presented guidelines on the circumstances in which each method should be used.

In this study, we were able to prove that the methodology of analyzing high-level features in the fully connected neural network and interpreting the knowledge inherent in the neural network with high-level features is valid. This paper is the first step to interpreting neural networks through features, and we will improve this methodology to apply to other models like convolutional neural networks and complex image domains in future work (Appendix A).

**Author Contributions:** Conceptualization, methodology, E.-H.L. and H.K.; validation, formal analysis, investigation, visualization, and writing—original draft preparation, E.-H.L.; writing—review and editing, supervision, project administration, and funding acquisition, H.K.; All authors have read and agreed to the published version of the manuscript.

**Funding:** This work was supported by Institute for Information & communications Technology Planning & Evaluation (IITP) grant funded by the Korea government (MSIT) (No. 2020-0-00368, A Neural-Symbolic Model for Knowledge Acquisition and Inference Techniques).

**Institutional Review Board Statement:** Not applicable.

**Informed Consent Statement:** Not applicable.

**Data Availability Statement:** Publicly available datasets were analyzed in this study. This data can be found here: MNIST dataset [28] at http://yann.lecun.com/exdb/mnist/, notMNIST dataset [29] at https://yaroslavvb.blogspot.it/2011/09/notmnist-dataset.html, and fashion-MNIST dataset [30] at https://github.com/zalandoresearch/fashion-mnist.

**Conflicts of Interest:** The authors declare no conflict of interest.

**Abbreviations**

The following abbreviations are used in this manuscript:

| | |
|---|---|
| XAI | Explainable Artificial Intelligence |
| DNF | Disjunctive Normal Form |
| OAS | Ordered-Attribute Search |
| CRED | Continuous/discrete Rule Extractor via Decision tree induction |
| FEB-RE | Feature-based Rule Explanation |
| ReLU | Rectified Linear Unit |
| ELU | Exponential Linear Unit |

**Appendix A. Explanations on Full Dataset**

We present the generated explanations by FEB-RE with the neural network trained on a full dataset in Figure A1. Figure A1a shows activation maps of high-level features of neural networks trained with the notMNIST full dataset, and Figure A1b represents some of the generated explanations for each class of notMNIST.

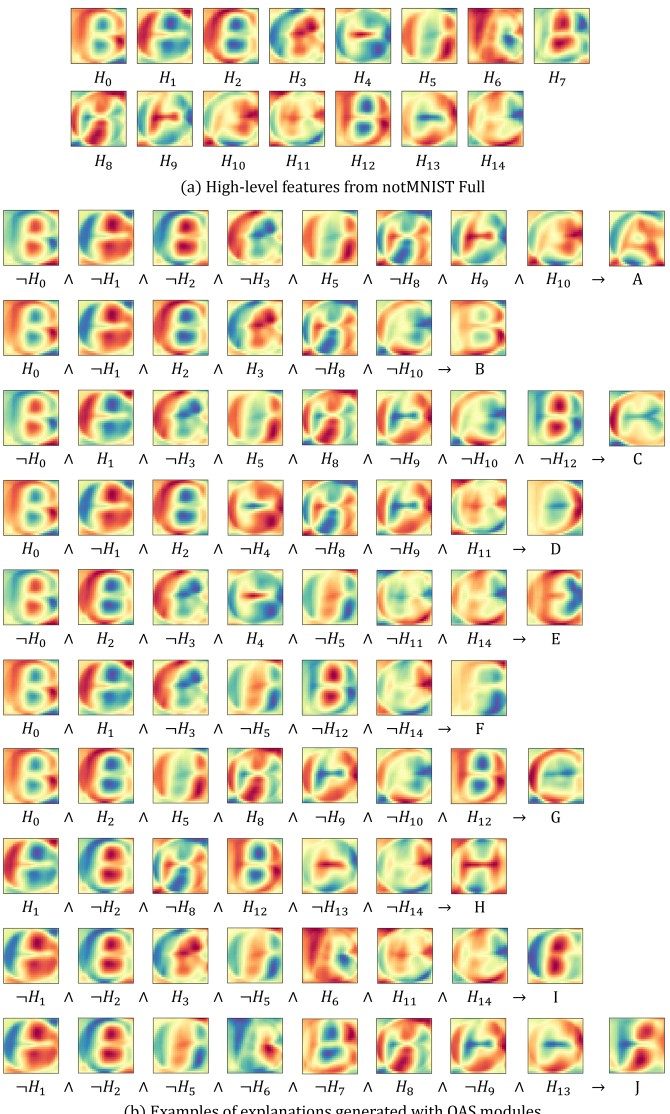

(a) High-level features from notMNIST Full

(b) Examples of explanations generated with OAS modules

**Figure A1.** (**a**) Activation maps of the high-level features after adapt pruning techniques with the neural network trained by notMNIST Full dataset. (**b**) Sample of the explanations for each class in the notMNIST Full dataset generated by FEB-RE with OAS module.

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
