# Peer review of "Feature-Based Interpretation of the Deep Neural Network"

_electronics, doi:10.3390/electronics10212687_

Round 1

Reviewer 1 Report

General
-------

The authors introduce a method for a feature-based interpretation of deep neural
networks. Their proposed FEB-RE method generates a “global explanation based on
high-level features of the neural network”. To be more specific, the method
trains a fully connected network, prunes a selected hidden layer and extracts a
set of if-then rules.

While the method seems to be reasonable for structured data, the exemplary
selected image domain seems to be wrong. Image processing with fully connected
neural networks is only applicable for low-resolution images.

The authors also claim that the method offers a human-friendly explanation for
the entire network while also state, that “trained high-level features can be
complex for humans to understand when using full datasets”. As a result, they
only use four out of ten classes of the examined datasets.

In addition, for the evaluation, the authors use the train dataset for the
accuracy and fidelity to measure how well the network explains the trained
knowledge. This approach heavily favors overfitted networks.

Content
-------

The document contains some contradictory statements
— Page 4/5, line 197/198: “…the high-level feature layer is limited to using
only the sigmoid or tanh activation functions.” and Page 7, line 254/255:
“High-level feature layer use sigmoid function as activation function, so
0 < h(x) < 1”. tanh(x) is in [-1,1].

— Page 4, line 180: “…human-friendly explanation …” and Page 8, line
321/322: “… high-level features can be complex for humans to understand
when using full datasets.”

— Page 13, line 482/483: “There seems to be no reason to use the OAS module
because the CRED module is superior in all aspects” and Page 14, line
507/508: “For all datasets, the explanation generated from the OAS module
has higher fidelity.”

What exactly do you mean with: Page 14, line 502-506?

Grammar and Language
--------------------

The authors should carefully read the document again and correct the grammar.
The first paragraph of section 3.1 needs to be revised.

Conclusion
----------

Despite several grammar mistakes, the paper is written comprehensible. As
mentioned in the general section, the applicability of the method in the image
domain is questionable. This, combined with the contradictory statements and the
overall weaknesses of the method lead to a reject.

Author Response

Dear reviewer.

Thank you for your detailed comments on our work. We have checked your feedback carefully and have been able to revise the paper based on it. The followings are the answer to your comment.

Point 1. While the method seems to be reasonable for structured data, the exemplary selected image domain seems to be wrong. Image processing with fully connected neural networks is only applicable for low-resolution images.

Response 1. Our research focuses on interpreting neural networks through high-level features rather than providing an explanation for images. In this study, we were able to prove that the methodology of analyzing high-level features in the fully connected neural network and interpreting the knowledge inherent in the neural network with high-level features is valid. This paper is the first step to interpret neural networks through features, and we will improve this methodology to apply to other models like convolutional neural networks and complex image domains in future work.

In Section 5.3, we provide a qualitative analysis of the proposed FEB-RE and previous works that provide an explanation for CNN. Through this, we think it is able to address the difference between the existing XAI technique and our proposed explanation technique to represent the differentiation of our FEB-RE method.

Point 2. The authors also claim that the method offers a human-friendly explanation for the entire network while also state, that “trained high-level features can be complex for humans to understand when using full datasets”. As a result, they only use four out of ten classes of the examined datasets.

Response 2. Because the term 'human-friendly is too ambiguous, we describe it in detail. We chose four classes because the premise is long, and it isn't easy to include all the contents as a figure in the main body. The explanation generated for the full dataset using all ten classes is supplemented in the Appendix A.

Point 3. In addition, for the evaluation, the authors use the train dataset for the accuracy and fidelity to measure how well the network explains the trained knowledge. This approach heavily favors overfitted networks.

Response 3. Our explanation method focuses on expressing the knowledge inherent in the trained neural network. Therefore, we calculated the accuracy and fidelity using the training dataset used to train the neural network.

Our method may produces a better explanation in the overfitted network, as you said, but as mentioned in Section 4.2, we apply early stopping during training, and the test accuracy in Table 3 shows that the overfitted networks were not used during the experiment.

Point 4. “…the high-level feature layer is limited to using only the sigmoid or tanh activation functions.” and Page 7, line 254/255: “High-level feature layer use sigmoid function as activation function, so 0 < h(x) < 1”

Response 4. The tanh function can be used with slight modifications, but we remove tanh in paper because the thesis of the paper is generally written based on the sigmoid function.

Point 5. “There seems to be no reason to use the OAS module because the CRED module is superior in all aspects” and Page 14, line 507/508: “For all datasets, the explanation generated from the OAS module has higher fidelity.”

Response 5. These two sentences describe the results of different experiments, but we have written vaguely. We rephrased the paragraph to deliver the intended meaning.

Reviewer 2 Report

The reviewer appreciates the authors efforts in providing a convincing interpretation of a Deep Nueral Network. The paper is well-written and good to follow. However, I have the following concerns which can be adressed in a major revision:

a) How can you link this interpretation with other FNN variants like RFF-DL?

b) How do you know that the similarity in (5) is an accurately describes "similarity" sufficiently......you could have perhaps used higher dimensional similarity measures like correntropy etc.

c) How is your approach different from CNNs, which also does a multilevel/hierearchical description? And is relating it to CNN relevant?

d) Can your approach be hyperparameter-independent, i.e., work for all cases irresepective of hidden node-sizes and other parameters that are generally tuned in a scenario-dependent/dataset-dependent manner for DL.

While your theory seems interesting, perhaps you can generalize it along these lines and present a revised manuscript. 

I look forward to receiving the revised version.

Author Response

Dear reviewer.

Thank you for your detailed comments on our work. We have checked your feedback carefully and have been able to revise the paper based on it. The followings are the answer to your comment.

Point 1. How can you link this interpretation with other FNN variants like RFF-DL?

Response 1. The FEB-RE method we proposed can be used for the variant fully connected neural networks, which satisfies the limitations mentioned in Section 3.1. Because the interpretation of high-level features uses a pedagogical approach, it is not affected by the neural network structure. In the case of the RFF-DL you mentioned, if the high-level features layer for interpretation and the output layer are fully connected, there is no problem using the kernel in the hidden layer below it. Since RFF-DL creates a better decision boundary with the kernel, FEB-RE probably generates more refined explanations when applied to RFF-DL.

Point 2. How do you know that the similarity in (5) is an accurately describes "similarity" sufficiently......you could have perhaps used higher dimensional similarity measures like correntropy etc.

Response 2. We calculated the similarity between high-level features using Euclidean distance similarity to prune neural networks. Since the image domain used in our experiments is limited, network pruning was possible even with a simple similarity function. Application to complex data domains may need to use the complex similarity function, as you mentioned. We describe in more detail in Section 4.2 and Table 2 to show how the similarity function we used was appropriate and how we set the pruning parameter values.

Point 3. How is your approach different from CNNs, which also does a multilevel/hierearchical description? And is relating it to CNN relevant?

Response 3. Our proposed high-level feature analysis method is based on the correlation between global input and hidden units. Our approach is suitable for a fully connected neural network in which all input units are connected to the hidden unit, but since CNN's hidden unit(or feature map) focuses on the local input area, it is difficult to apply our method without modification. This paper is the first step to interpret neural networks through features, and we will improve this methodology to apply to other models like CNNs and complex image domains in future work.

In Section 5.3, we provide a qualitative analysis of the proposed FEB-RE and previous works that provide an explanation for CNN. Through this, we think it is able to address the difference between the existing XAI technique and our proposed explanation technique to represent the differentiation of our FEB-RE method.

Point 4. Can your approach be hyperparameter-independent, i.e., work for all cases irresepective of hidden node-sizes and other parameters that are generally tuned in a scenario-dependent/dataset-dependent manner for DL.

Response 4. Although we represent the experiment results using only two neural network structures in the paper, our method is hyperparameter-independent. The only hyperparameter that our FEB-RE approach limited is the activation function used in the hidden layer between the high-level feature layer and the output layer as mentioned in Section 3.1.

Reviewer 3 Report

The authors, within the manuscript, are proposing a framework for low level explainable CNNs. The paper is well structured and provide all the necessary details for explaining the suggested techniques. The main novelty of the work is the introduction a new approach for mapping the learning process of a deep network.

Bearing these in mind, I would suggest a minor revision of the manuscript, in which the authors will have the opportunity to address the following remarks.

  1. More details should be provided for the selection of threshold τr (l. 261).
  2. Figure 1 seems too generic. Please elaborate.
  3. A comparison with other eclectic xAI approach should be performed, in order for the reader to understand the gain the proposed model offers to the CNN final user. This would be my major comment.

    4. Section 6 should be more extend, as is seems unbalanced with the rest of the sections.

Author Response

Dear reviewer.

Thank you for your detailed comments on our work. We have checked your feedback carefully and have been able to revise the paper based on it. The followings are the answer to your comment.

Point 1. More details should be provided for the selection of threshold τr (l. 261).

Response 1. In Section 4.2, we added the process of determining the pruning parameter τr and added Table 2 to show the experimental results supporting this.

Point 2. Figure 1 seems too generic. Please elaborate.

Response 2. We add the byproduct and final product from each process in Figure 1 to make it easy to understand the role and output of each process.

Point 3. A comparison with other eclectic xAI approach should be performed, in order for the reader to understand the gain the proposed model offers to the CNN final user.

Response 3. Through Section 5.3 and Figure 5, we present a qualitative analysis of the previous XAI approaches(CAM, LIME, and LRP) with the proposed FEB-RE method. Through qualitative analysis, we think that the advantages of our proposed method compared to previous work can be demonstrated.

Point 4. Section 6 should be more extend, as is seems unbalanced with the rest of the sections.

Response 4. In the conclusion Section 6, we summarize the experimental results to support the paper's contribtion.

Round 2

Reviewer 1 Report

To my understanding (and according your conclusion) the one and only contribution is "FEB-RE". Thus, what is the abstract about? 

Author Response

Point1.

To my understanding (and according your conclusion) the one and only contribution is "FEB-RE". Thus, what is the abstract about?

Response 1.

The FEB-RE we proposed is a method that focuses on the units of the higher layer of the neural network, a high-level feature. With the feature-based approaches in the FEB-RE method, we show it is possible to generate global explanations from a deep neural network model trained with unstructured data.

By rephrasing lines 7~11 of the abstract, we revised it to reflect the basic idea of the proposed method better. Also, we corrected sentences in Section 1~Section 4 grammatically or semantically.

Reviewer 2 Report

The authors seem to have addressed the comments accurately, and the document can be accepted.

Author Response

We revised abstract it to reflect the basic idea of the proposed method better. Also, we corrected sentences in Section 1~Section 4 grammatically or semantically.